# Coarse-to-fine Knowledge Graph Domain Adaptation based on Distantly-supervised Iterative Training

## Abstract

Modern supervised learning neural network models require a large amount of manually labeled data, which makes the construction of domain-specific knowledge graphs time-consuming and labor-intensive. In parallel, although there has been much research on named entity recognition and relation extraction based on distantly supervised learning, constructing a domain-specific knowledge graph from large collections of textual data without manual annotations is still an urgent problem to be solved. In response, we propose an integrated framework for adapting and re-learning knowledge graphs from one coarse domain (biomedical) to a finer-define domain (oncology). In this framework, we apply distant-supervision on cross-domain knowledge graph adaptation. Consequently, no manual data annotation is required to train the model. We introduce a novel iterative training strategy to facilitate the discovery of domain-specific named entities and triples. Experimental results indicate that the proposed framework can perform domain adaptation and construction of knowledge graph efficiently.

## 1 Introduction

The triples in the knowledge graph (KG) contain the relationships between various entities, providing rich semantic background knowledge for various natural language processing (NLP) tasks, such as natural language representation Liu et al. (2020), question answering Saxena et al. (2020), image captioning Zhang et al. (2021a), and text classification Jiang et al. (2020). Consequently, automatically constructing knowledge graphs directly from natural texts has attracted close attentions of re-searchers in recent years Kertkeidkachorn & Ichise (2017); Rossanez et al. (2020); Stewart & Liu (2020).

KG construction from text generally involves two primitive steps: named entity recognition (NER) and relation extraction (RE). Named entity recognition aims to identify the types of entities mentioned in text sequences, such as people, place, etc. in the open domain; or disease, medicine, disease symptom, etc. in the biomedical domain. The relation extraction also known as triple extraction, aims to identify the relationship between two entities, such as the birthplace relationship between people and places in the open domain; or the therapeutic relationship between drug and disease in the biomedical domain. NER and RE are necessary steps for information extraction to construct KG from the text. In addition to NER and RE, constructing a KG usually includes other steps such as coreference resolution, entity linking, knowledge fusion, and ontology extraction. In order to facilitate model evaluation, this paper mainly focuses on information extraction and then constructs a KG.

In the construction of fine-domain KG scenarios, there are usually some existing resources available, such as biomedical KGs in coarse domains, which generally cover broader concepts and more commonsense knowledge. When constructing the oncology KG, the biomedical KG is thus available. However, few studies have focused on adapting KG from the coarse domain (e.g., biomedical) to the fine domain (e.g., oncology) where a large collection of unlabeled textual data are available, which motivates the work in this paper.

Distant supervision Smirnova & Cudré-Mauroux (2018) is an intuitive way to transfer coarse-domain KG to fine domains. Distant-supervision provides labels for data with the help of an external

knowledge base, which saves the time of manual labeling. For distantly-supervised NER, we can build distant labels by matching unlabeled sentences with external semantic dictionaries or knowledge bases. The matching strategies usually include string matching Zhao et al. (2019), regular expressions Fries et al. (2017), and some heuristic rules. The distantly-supervised RE holds an assumption Mintz et al. (2009): if two entities participate in a relation, then any sentence that contains those two entities might express that relation. Following this assumption, any sentence mentioning a pair of entities that have a relation according to the knowledge base will be labeled with this relation Smirnova & Cudré-Mauroux (2018).

Therefore, the KG in the coarse domain can be potentially used as a knowledge base for distant supervision, thus avoiding a large number of manual annotations. However, only using the KG of the coarse domain as the knowledge base might limit the model's ability to discover domain-specific named entities and triples in the fine domain, which further limits the construction of the fine domain KG. To address these problems, in this paper, we propose a novel coarse-to-fine knowledge graph domain adaptation (KGDA) framework. Our KGDA framework utilizes an iterative training strategy to enhance the model's ability to discover fine-domain entities and triples, thereby facilitating fast and effective coarse-to-fine KG domain adaptation.

Overall, the contributions of our work are as follows:

- An integrated framework for adapting and re-learning KG from coarse-domain to fine-domain is proposed. As a case study, the biomedical domain and oncology domain are considered the coarse domain and fine domain, respectively.

- Our model does not require human annotated samples with distant-supervision for cross-domain KG adaptation, and the iterative training strategy is applied to discovering domain-specific named entities and new triples.

- The proposed method can be adapted to various pre-trained language models (PLMs) and can be easily applied to different coarse-to-fine KGDA tasks. It is so far the simplest data-driven approach for learning a KG from free text data, with the help of the coarse domain KG.

- Experimental results demonstrate the effectiveness of the proposed KGDA framework. We will release the source code and the data used in this work to fuel further research. The constructed oncology KG will be hosted as a web service to be used by the general public.

## 2 RELATED WORK

### 2.1 PIPELINE-BASED METHODS FOR KG CONSTRUCTION

The pipeline-based methods apply carefully-crafted linguistic and statistical patterns to extract the co-occurred noun phrases as triples. There are many off-the-shelf toolkits available, for example, Stanford CoreNLP Manning et al. (2014), NLTK Thanaki (2017), and spaCy, which can be used for the NER tasks; Reveb Fader et al. (2011), OLLIE Schmitz et al. (2012), and Stanford OpenIE Angeli et al. (2015) can be used for the information extraction task. There have been multiple pipelines Mehta et al. (2019); Rossanez et al. (2020) developed as well, consisting of modules targeting different functionalities needed for the KG construction. However, the pre-defined rules of off-the-shelf toolkits are generally tailored to specific domains, such methods are not domain-agnostic, and a new set of rules will be needed for a new domain.

### 2.2 DATA-DRIVEN METHODS FOR KG CONSTRUCTION

With the development of representation learning in language models, researchers began to apply data-driven models to solve the KG construction tasks. Based on how the model is trained, these works can be divided into three categories: fully-supervised methods Zhao et al. (2019); Li et al. (2022b), semi-supervised methods Zahera et al. (2021), and weakly-supervised methods Yu et al. (2021). We will introduce the methods of fully-supervised and weakly-supervised in this section. Specifically, the NER, RE, and entity linking tasks in the KG construction pipeline can all be solved by fully-supervised learning methods such as long short-term memory neural network (LSTM) Hochreiter & Schmidhuber (1997); Zeng et al. (2017). Graph neural network methods have also

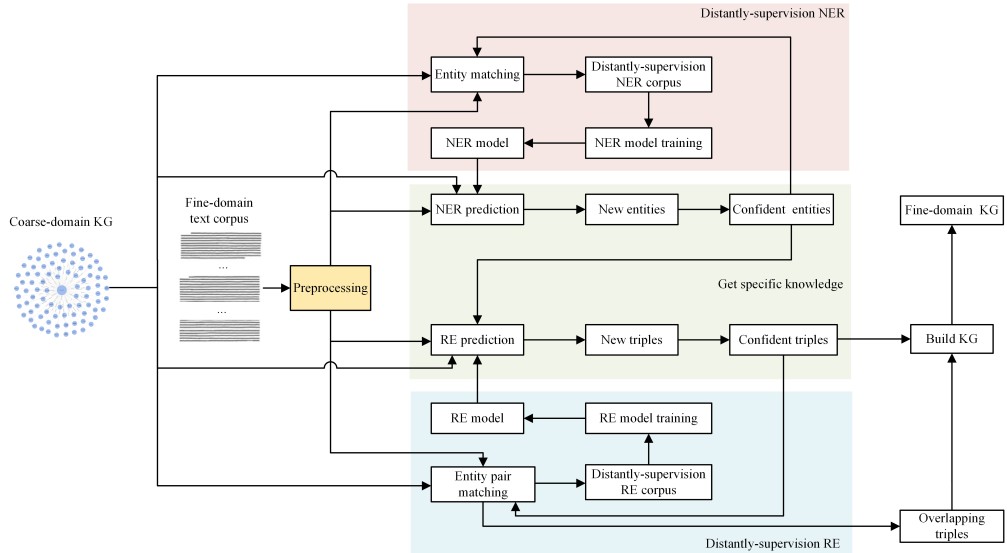

Figure 1: The overall framework of iterative training KGDA.

been applied for domain-specific NER tasks Chen et al. (2021) and document-level RE Zhang et al. (2020). The bidirectional encoder representation from transformers (BERT) Kenton & Toutanova (2019), the widely-used PLM, can also tackle the NER Jia et al. (2020), RE Roy & Pan (2021), and entity linking Li et al. (2022a) tasks. While the advancement of deep learning-based methods has greatly improved the effectiveness of KG construction, fully-supervised learning requires a large amount of human-annotated data text. Furthermore, the annotation can only be domain-specific, making it difficult to transfer the KG construction work to a new domain, and ultimately limiting the scalability and efficiency of the research in KG.

On the other hand, distant supervision, a weakly supervised learning method, can replace manual annotation with an existing and remote knowledge base. Previous studies have applied remote supervised learning to deal with NER Zheng et al. (2021), and RE Wei (2021); Zhang et al. (2021b) tasks. Thus in this work, we adopted the distant-supervision scheme in the proposed KGDA framework. It should be noted that KG of the coarse domain (e.g., biomedical) generally will not contain the complete knowledge of its finer sub-domains (e.g., oncology). So when we use the coarse-domain KG for distant supervision, labels of the target domain will be limited by the source domain, making it less effective to discover new knowledge. To address this issue, we introduced an iterative strategy to gradually update the model via distant supervision while at the same time using the partially-trained model to discover new entities and relations from the data of the target fine domain.

## 3 METHODOLOGY

### 3.1 NOTATION AND TASK DEFINITION

An unstructured sentence $s = [w_1, w_2, w_3, ..., w_n]$ indicates a sequence of tokens, where $n$ is its length. A dataset $\mathbb{D}$ is a collection of unstructured sentences (i.e. $\mathbb{D} = \{s_1, s_2, s_3, .., s_m\}$). The knowledge graph, denoted as $\mathbb{K}$, is a collection of triples $t = (e_i, r_j, e_k)$, where $e_i \in \mathbb{E}$ and $e_k \in \mathbb{E}$ are the head entity and the tail entity respectively, and $r_j \in \mathbb{V}$ is the relation between $e_i$ and $e_k$. Here we denote coarse-domain KG as $\mathbb{K}_c$ and fine-domain KG as $\mathbb{K}_f$.

In a typical scenario of KG domain adaptation, we will have an existing coarse-domain KG and a large amount of unlabeled text in the fine domain. For example, when constructing the oncology KG, we can utilize the existing biomedical KG and collect oncology-related literature as unlabeled text. KG constructed from the fine domain data would then include overlapping triples with the coarse-domain KG and new triples representing domain-specific knowledge. Specifically, the fine-domain KG contains the following three types of triples:

- **Overlapping triples** $\mathbb{T}_O$: Triples that also existed in the coarse-domain KG, indicating knowledge overlapping between the coarse and fine domains.

- **Triples of new relations but overlapping entities** $\mathbb{T}_R$: Triples with both entity pairs existing in the coarse-domain KG but no indicated relationships between these entity pairs.

- **Triples of new entities** $\mathbb{T}_E$: Triples with at least one entity not existing in the coarse-domain KG. Consequently, the relationship is also unknown in the coarse domain.

Both $\mathbb{T}_R$ and $\mathbb{T}_E$ belong to the specific knowledge of the fine domain. The goal of the coarse-to-fine KGDA task is to adapt the KG from the coarse domain to the fine domain and leverage the knowledge from the coarse domain to guide the mining of new knowledge specific to the fine domain. Finally, we will keep the definition of entity types and relation types from coarse-domain KG when constructing the fine-domain KG.

## 3.2 ITERATIVE TRAINING FRAMEWORK

While it is trivial to identify the overlapping entities $\mathbb{E}_O$ and triples $\mathbb{T}_O$ by distant supervision, if the NER and RE models are trained on the entire corpus, they will not be able to recognize the fine domain-specific named entities and triples ($\mathbb{T}_R$ and $\mathbb{T}_E$). Because the distant-supervision labels are generated by matching $\mathbb{K}_c$. Thus we introduce an iterative training strategy to construct $\mathbb{T}_R$ and $\mathbb{T}_E$ from the text and adapt the knowledge from $\mathbb{K}_c$ to $\mathbb{K}_f$.

The overall framework of the iterative training scheme is shown in Fig. 1, and the detailed pseudo code can be found in Algorithm 1. Rather than performing distant-supervision training on the whole unlabeled text corpus, the core mechanism of the proposed iterative training is to split the whole unlabeled dataset into $n$ sub-datasets without intersection. Before building distant-supervision corpus, the trained model is used to predict the text corpus for getting specific knowledge of fine-domain, which is conducive to mining $\mathbb{T}_R$ and $\mathbb{T}_E$ of the fine-domain.

As shown in Figure 1, firstly, it is necessary to preprocess the acquired text corpus in the fine domain. Preprocessing operations include: handling special characters, word segmentation, filtering sentences using human-defined rules (such as sentence length), etc. Then, our framework involves two neural network models: NER model and RE model. We replace the PLM's output layer with a classifier head as NER model $model_N$ and fine-tune it by minimizing the cross-entropy loss on distant-supervision NER corpus. Additionally, we apply the BIO scheme Li et al. (2012) to generate NER sequence labels. For the RE task, we use the template to generate distant-supervision samples. The template we adopted is "[CLS] *head entity* (*head entity type*) [SEP] *tail entity* (*tail entity type*) [SEP] *sentence*". The RE model $mode_R$ is defined as a PLM with a fully connected layer as a relation classifier. The feature of special token [CLS] fed into this fully connected layer and fine-tune $mode_R$ by minimizing the cross-entropy loss on distant-supervision RE corpus.

We summarize the steps to achieve KGDA in Algorithm 1. For the first parts of the text corpus $\mathbb{D}_1$, the distant-supervision method is applied to construct the NER training corpus $corp_N$ and RE training corpus $corp_R$, and the NER model $model_N$ and RE model $mode_R$ are trained based on corpus $corp_N$ and $corp_R$, respectively. For other part of the text corpus $\mathbb{D}_i$, we apply the previously trained $model_N$ and $mode_R$ to extract the entities and triples in the fine-domain, and select the high confidence entities $\mathbb{E}_{conf}$ and high confidence triples $\mathbb{T}_{conf}$ as the specific knowledge of the fine-domain (line 7). Then, we take $\mathbb{K}_c$, $\mathbb{E}_{conf}$, and $\mathbb{T}_{conf}$ as the external knowledge base for constructing distant-supervision $corp_N$ and $corp_R$ (line 8). Finally, we use overlapping triples $\mathbb{T}_O$ and high-confidence triples $\mathbb{T}_{conf}$ to construct a knowledge graph of fine domains (line 17).

Next, we show the details of *get_distant_corpus* in Algorithm 2 and *get_specific_ knowledge* in Algorithm 3.

## 3.3 CONSTRUCTING DISTANTLY-SUPERVISED CORPUS

Through distant-supervision, we can only match entity pairs that have a relationship and use them as positive samples. We then construct negative samples with NULL relationship by the following two schemes: 1) randomly sampling two entities which have no relationship as defined in the coarse-domain; 2) randomly sampling a word from out-of-domain words (i.e., a word that is not an entity as defined in the coarse domain) $\mathbb{W}_O$ as one of the entities. The parameter $ratio_n$ controls the ratio

---

**Algorithm 1** Iterative training KGDA framework

**Input**: Text corpus $\mathbb{D} = \{\mathbb{D}_1, \mathbb{D}_2, ..., \mathbb{D}_n\}$, coarse-domain KG $\mathbb{K}_c$, out-of-domain words $\mathbb{W}_O$
**Parameter**: Initialized NER model $model_N$, initialized RE model $model_R$
**Output**: fine-domain kg $\mathbb{K}_f$

1: Let new entities $\mathbb{E}_{new} = \{\}$ , new entities with high confidence $\mathbb{E}_{conf} = \{\}$ , new triples $\mathbb{T}_{new} = \{\}$ , new triples with high confidence $\mathbb{T}_{conf} = \{\}$ .
2: $corp_N, corp_R, \mathbb{E}_O, \mathbb{T}_O$= build_distant_corpus( $\mathbb{D}_1, \mathbb{K}_c, \mathbb{E}_{conf}, \mathbb{T}_{conf}, \mathbb{W}_O$ )
3: train_NER($model_N, corp_N$)
4: train_RE($model_R, corp_R$)
5: $i = 2$
6: **while** $i <= n$ **do**
7:    $\mathbb{E}_{new}, \mathbb{E}_{conf}, \mathbb{T}_{new}, \mathbb{T}_{conf}$ = get_specific_knowledge($\mathbb{D}_i, \mathbb{K}_c, \mathbb{E}_{new}, \mathbb{E}_{conf}, \mathbb{T}_{new}, \mathbb{T}_{conf}$ )
8:    $corp'_N, corp'_R, \mathbb{E}'_O, \mathbb{T}'_O$= get_distant_corpus( $\mathbb{D}_i, \mathbb{K}_c, \mathbb{E}_{conf}, \mathbb{T}_{conf}, \mathbb{W}_O$)
9:    $corp_N = corp_N \cup corp'_N$
10:   $corp_R = corp_R \cup corp'_R$
11:   $\mathbb{E}_O = \mathbb{E}_O \cup \mathbb{E}'_O$
12:   $\mathbb{T}_O = \mathbb{T}_O \cup \mathbb{T}'_O$
13:   train_NER($model_N, corp_N$)
14:   train_RE($model_R, corp_R$)
15:   $i = i + 1$
16: **end while**
17: $\mathbb{K}_f$ = build_kg($\mathbb{T}_O$ , $\mathbb{T}_{conf}$)
18: **return** $\mathbb{K}_f$

---

**Algorithm 2** Constructing distantly-supervised corpus

**Input**: A part of text corpus text corpus $\mathbb{D}_i$, coarse-domain KG $\mathbb{K}_c$, new entities with high confidence $\mathbb{E}_{conf}$, new triples with high confidence $\mathbb{T}_{conf}$, out-of-domain words $\mathbb{W}_O$
**Parameter**: negative sample ratio $ratio_n$ , out-of-domain sample ratio $ratio_o$
**Output**: Distant-supervision NER corpus $corp_N$, distant-supervision RE corpus $corp_R$, overlapping entities $\mathbb{E}_O$, overlapping triples $\mathbb{T}_O$

1: Let $corp_E = \{\}, corp_R = \{\}, \mathbb{E}_O = \{\}, \mathbb{T}_O = \{\}$.
2: sentence_num = len( $\mathbb{D}_i$)
3: $j = 1$
4: **while** j<=sentence_num **do**
5:   $entities$ = entity_matching( $\mathbb{D}_i^j, \mathbb{K}_c, \mathbb{E}_{conf}$)
6:   $\mathbb{E}_O = \mathbb{E}_O \cup entities$
7:   $corp_N = corp_N \cup$ build_NER_sample( $\mathbb{D}_i^j, entities$)
8:   $triples_k, triples_c$ = entity_pair_matching( $\mathbb{D}_i^j, \mathbb{K}_c, \mathbb{T}_{conf}$)
9:   $triples = triples_k \cup triples_c$
10:   $triples_n$ = get_negative_triples( $\mathbb{D}_i^j, \mathbb{W}_O, triples, ratio_n, ratio_o$)
11:   $corp_R = corp_R \cup$ get_samples($triples$)
12:   $corp_R = corp_R \cup$ get_samples($triples_n$)
13:   $\mathbb{T}_O = \mathbb{T}_O \cup triples_k$
14:   $j = j + 1$
15: **end while**
16: **return** $corp_N, corp_R, \mathbb{E}_O, \mathbb{T}_O$

---

of negative samples (constructed by either schemes) to the total sample size. The parameter $ratio_o$ controls the ratio of entity pairs constructed by the second scheme (i.e., via sampling the words outside the domain) to the size of negative samples, respectively.

In addition to the $\mathbb{K}_c$ in the source domain, we use both $\mathbb{K}_c, \mathbb{E}_{conf}$, and $\mathbb{T}_{conf}$ as knowledge bases for constructing the remotely supervised corpus. This would ensure that the NER and RE models can identify the overlapping knowledge between $\mathbb{K}_c$ and $\mathbb{K}_f$, while at the same time be guided to discover the new knowledge specific to the fine domain.

---

**Algorithm 3** Discovering fine-domain specific knowledge

---

**Input**: A part of text corpus text corpus $\mathbb{D}_i$, coarse-domain KG $\mathbb{K}_c$, new entities $\mathbb{E}_{new}$, new entities with high confidence $\mathbb{E}_{conf}$, new triples $\mathbb{T}_{new}$, new triples with high confidence $\mathbb{T}_{conf}$
**Parameter**: NER model $model_N$, RE model $model_R$, probability threshold of the entity $th_{pe}$, frequency threshold of the entity $th_{fe}$, probability threshold of the triple $th_{pt}$, frequency threshold of the triple $th_{ft}$
**Output**: $\mathbb{E}_{new}, \mathbb{E}_{conf}, \mathbb{T}_{new}, \mathbb{T}_{conf}$

  1: Let $corp_E = \{\}, corp_R = \{\}, \mathbb{E}_O = \{\}, \mathbb{T}_O = \{\}$.
  2: sentence_num = len( $\mathbb{D}_i$ )
  3: $j = 1$
  4: **while** $j <= sentence\_num$ **do**
  5:     $entities$ = NER_prediction( $\mathbb{D}_i^j, model_N$ )
  6:     $entities$ = get_new_entities( $entities, \mathbb{K}_c$ )
  7:     $\mathbb{E}_{new}$ = merge_entity( $\mathbb{E}_{new}, entities$ )
  8:     $j = j + 1$
  9: **end while**
 10: $\mathbb{E}_{conf}$ = get_confidence_entity( $\mathbb{E}_{new}, th_{pe}, th_{fe}$ )
 11: $j = 1$
 12: **while** $j <= sentence\_num$ **do**
 13:     $entities$ = entity_matching( $\mathbb{D}_i^j, \mathbb{K}_c, \mathbb{E}_{conf}$ )
 14:     $pairs$ = enumerate_pairs( $entities$ )
 15:     $pairs$ = get_new_pairs( $pairs, \mathbb{K}_c$ )
 16:     $triples$ = RE_prediction( $\mathbb{D}_i^j, pairs, model_R$ )
 17:     $\mathbb{T}_{new}$ = merge_triple( $\mathbb{T}_{new}, triples$ )
 18:     $j = j + 1$
 19: **end while**
 20: $\mathbb{T}_{conf}$ = get_confidence_triple( $\mathbb{T}_{new}, th_{pt}, th_{pt}$ )
 21: **return** $\mathbb{E}_{new}, \mathbb{E}_{conf}, \mathbb{T}_{new}, \mathbb{T}_{conf}$

---

As shown in Algorithm 2, for building the distantly-supervised NER corpus $corp_N$, the sentence $\mathbb{D}_i^j$ is firstly string-matched with the knowledge bases $\mathbb{K}_c$ and $\mathbb{E}_{conf}$ to extract the entities in the sentence (line 5). Afterward, the matched entities are merged into overlapping entities $\mathbb{E}_O$, and the NER label sequences are generated through the BIO strategy to merge into $corp_N$ (line 6 and 7). For building the distantly-supervised RE corpus $corp_R$, we firstly take $\mathbb{K}_c$ and $\mathbb{T}_{conf}$ as knowledge bases and use entity pair matching to match the triples $triples_k$ based on $\mathbb{K}_c$ and the triples $triples_c$ based on $\mathbb{T}_{conf}$ appearing in the sentence $\mathbb{D}_i^j$ (line 8). We then build negative triples with parameters $ratio_n$ and $ratio_o$ (line 10). Finally, we construct the RE corpus based on the triples $triples$, $triples_n$ and corresponding sentences through a pre-defined relationship sample template (line 11 and 12).

## 3.4 DISCOVERING FINE-DOMAIN SPECIFIC KNOWLEDGE

Recall that in the proposed iterative training framework, the whole unlabeled dataset is divided into $n$ sub-dataset $\mathbb{D}_i, i = 1...n$, the fine-domain specific knowledge discovery will be performed on each sub-dataset except the first one $\mathbb{D}_i, i = 2...n$ (line 5 to 16 in Algorithm 1). For each new sub-dataset $\mathbb{D}_i, i = 2...n$, we will use the previously-updated models $model_N$ and $model_R$ to predict the new entities and triples. Afterward, the sub-dataset will be used for updating $model_N$ and $model_R$ via distantly-supervised training. As noisy or incorrect entities and triples could be discovered during this procedure, we developed a filtering mechanism only to keep the entities and triples with higher confidence. Specifically, we design the rules for filtering the discovered entities and triples by: 1) probability of the new entities and triples predicted by the corresponding models should be greater than pre-defined thresholds $th_{pe}$ and $th_{pt}$, respectively; 2) cumulative frequency of the new entities and triples discovered from datasets $\mathbb{D}_2$ to $\mathbb{D}_i$ should be greater than the pre-defined thresholds $th_{fe}$ and $th_{ft}$, respectively.

As shown in Algorithm 3, for discovering new entities $\mathbb{E}_{new}$, we will apply the trained $model_N$ on dataset $\mathbb{D}_i$ and obtain $entities$ that are disjoint with $\mathbb{K}_c$ (line 5 and 6). Then, we will merge $entities$ with the previously-discovered entity set $\mathbb{E}_{new}$ (line 7). Finally, we will select the "high-confident"

entity as $\mathbb{E}_{conf}$ based on the mechanism above by the prediction probability and cumulative frequency (line 10). For the discovery of new triples $\mathbb{T}_{new}$, we will enumerate entity pairs that are disjoint with the $\mathbb{K}_c$ (line 13 - 15). We will then use the trained RE model and the predefined sample template to predict the relationship of the entity pairs and delete the triples whose predicted relationship is NULL (line 16). Other processing is similar to the discovery of new entities.

After Algorithm 3, discovered entities specific to the fine domain are stored in $\mathbb{E}_{conf}$. Discovered triples $\mathbb{T}_R$ (new relation, overlapping entity) and $\mathbb{T}_E$ (new relation, new entity) are stored in $\mathbb{T}_{conf}$. In the next iteration, Algorithm 2 will then use the updated $\mathbb{E}_{conf}$ and $\mathbb{T}_{conf}$ for building distant-supervision corpus. Such iterative design can facilitate the interoperability between the two competing tasks based on a fixed number of unannotated data samples in the fine target domain: distantly-supervised training of the NER and RE models versus the discovery of new knowledge using the trained NER and RE models, thus improve the efficiency of performing KG domain adaptation and construction without any annotation.

## 4 EXPERIMENTS

In this work, we used the adaptation of KG from the biomedical domain (coarse) to the oncology domain (fine) as an example to demonstrate the workflow of the KGDA framework, as well as to evaluate its effectiveness in practice. Implementation details of the experiment are also provided, along with the publicly-available data and the containerized environment in the released source code, for easy replication of the experiment and the development of other KG methods.

### 4.1 DATASET

We downloaded papers from 12 international journals (journal details can be found in the supplemental materials) in the oncology domain. PDF files of the papers were cleaned and converted to sentences. In total, we included nearly 240,000 sentences as the unlabeled text corpus of the oncology domain $\mathbb{D}$. The coarse-domain KG $\mathbb{K}_c$ used in this work is the biomedical KG[1], defines 18 entity types and 19 relationship types, including 5.2 million English entities and 7.34 million triples. The lists of entity types and relationship types can be found in the supplementary materials.

### 4.2 EVALUATION

Similar to the previous works Mintz et al. (2009), we evaluate our method in two schemes: held-out evaluation and manual evaluation. For the held-out evaluation, we reserved a part of the text corpus of $\mathbb{D}$ as the test set. During the testing, we then compared the prediction results of the NER and RE models with the labels matched with $\mathbb{K}_c$, and calculated the precision, recall, and F1 of the held-out dataset. Specifically, we use seqvel[2] to evaluate the micro average precision, recall, F1 of NER. When evaluating the RE model, we perform relation classification prediction on the triples existing in $\mathbb{K}_c$ and corresponding entity pairs appearing in the held-out corpus. Finally, weighted average precision, recall, and F1 from the held-out evaluation will be reported.

As the labels of testing samples in the held-out evaluation are all inferred by distant supervision from the coarse domain, such scheme can only evaluate whether the trained model can capture the knowledge in the coarse domain, but cannot evaluate the ability of the models to discover new knowledge in the fine-domain. Therefore, we also adopted the manual evaluation scheme, consisting of the evaluations of: 1) the entities specific to fine domain $\mathbb{E}_{conf}$, which are not presented in $\mathbb{K}_c$; 2) the triples of new relations $\mathbb{T}_R$; 3) the triples of new entities $\mathbb{T}_E$. We randomly sampled 50 cases of $\mathbb{E}_{conf}$, $\mathbb{T}_R$, and $\mathbb{T}_E$ respectively, then asked one physician to manually label them for whether the entities and triples are correct. As the number of name entities and triples instances that are expressed in the corpus is unknown, we cannot estimate the recall of fine-domain KG. Therefore, we only show the precision of $\mathbb{E}_{conf}$, $\mathbb{T}_R$, and $\mathbb{T}_E$. We fully recognize that the discovery of new knowledge in the fine-domain is an indispensable task for this work and we are recruiting more medical experts to conduct human reader study and performance evaluation for the proposed model.

---

[1]https://idea.edu.cn/bios.html
[2]https://github.com/chakki-works/seqeval

### 4.3 IMPLEMENTATION SETTINGS

We divide the corpus $\mathbb{D}$ into six equal subsets, and each subset contains around 40,000 sentences. We used $\mathbb{D}_1$ to $\mathbb{D}_5$ for model training and KG construction. We reserved $\mathbb{D}_6$ for held-out evaluation. We tested BERT Kenton & Toutanova (2019), Bio_ClinicalBERT Alsentzer et al. (2019), biomed_RoBERTa Gururangan et al. (2020) for initializing NER and RE models. Our experiments were run on an Ubuntu system computer with three NVIDIA 1080Ti graphics cards. The learning rate, batch size, and epochs are set as 2E-05, 20, and 4, respectively. Hyperparameters $th_{fe}, th_{pe}, th_{ft}, th_{pt}$ are set as 2, 0.95, 3, 0.97. The parameters $ratio_n$ and $ratio_o$ that control negative sampling are set to 0.2 and 0.3.

### 4.4 HELD-OUT EVALUATION

Table 1: Held-out evaluation of NER model.

| models | precision | recall | F1 |
|---|---|---|---|
| BERT | **0.878** | **0.859** | **0.868** |
| Bio_ClinicalBERT | 0.877 | 0.853 | 0.865 |
| biomed_RoBERTa | 0.871 | 0.848 | 0.859 |

Table 2: Held-out evaluation of RE model.

| models | precision | recall | F1 |
|---|---|---|---|
| BERT | **0.977** | 0.912 | 0.943 |
| Bio_ClinicalBERT | 0.976 | 0.923 | 0.948 |
| biomed_RoBERTa | 0.973 | **0.932** | **0.952** |

The results of the NER and RE models evaluated by the held-out dataset are shown in Table 1 and Table 2, respectively. The KGDA frameworks initialized by the three pre-trained language models (BERT, Bio_ClinicalBERT, and biomed_RoBERTa) all show good performance in held-out evaluations, demonstrating the robustness of our framework.

### 4.5 MANUAL EVALUATION

| models | #$\mathbb{E}_O$ | #$\mathbb{T}_O$ | #$\mathbb{E}_{conf}$ | #$\mathbb{T}_R$ | #$\mathbb{T}_E$ |
|---|---|---|---|---|---|
| BERT | 69587 | 20615 | 1631 | 24580 | 905 |
| Bio_ClinicalBERT | 70055 | 20611 | 1936 | 25312 | 1195 |
| biomed_RoBERTa | 69497 | 20495 | 1625 | 23183 | 1071 |

Table 3: The number of entities and triples.

The number of all discovered entities ($\mathbb{E}_O$), triples ($\mathbb{T}_O$), new entities with high confidence ($\mathbb{E}_{conf}$), triples representing new relations with overlapping entities ($\mathbb{T}_R$), and triples representing new relations with new entities ($\mathbb{T}_E$) are shown in Table 3, with each row belonging to one pre-trained language models used. Numbers of $\mathbb{E}_O$ and $\mathbb{T}_O$ have minor differences among different pre-trained language models, possibly due to the conflicts in strings matching of knowledge bases. $\mathbb{E}_{conf}$, $\mathbb{T}_R$, and $\mathbb{T}_E$ represent specific knowledge of the fine domain. We sampled 50 cases from $\mathbb{E}_{conf}$, $\mathbb{T}_R$, and $\mathbb{T}_E$ for manual evaluation, and the results are shown in Table 4.

### 4.6 KNOWLEDGE GRAPH CONSTRUCTION IN THE FINE DOMAIN

As our ultimate goal, we can construct the KG in the fine domain by combining $\mathbb{T}_O$, $\mathbb{T}_R$, and $\mathbb{T}_E$. We selected biomed_RoBERTa as the backbone language model for KGDA and constructed the knowledge graph correspondingly. An example of the KG we built are shown in the supplementary material.

### 4.7 ABLATION STUDY

We investigated the impact of 3 techniques employed by KGDA on its held-out experiment performance by removing the corresponding component from the framework:

**w/o (cumulative)**: When using corpus $\mathbb{D}_i$ to train NER and RE models, the cumulative corpus is not used. i.e. delete lines 9 and 10 in Algorithm 1 and mark $corp'_N$ and $corp'_R$ in line 8 as $corp_N$ and $corp_R$ respectively.

**w/o (iter)**: Remove the iterative training strategy and only use $\mathbb{K}_C$ as an external knowledge base.

**w/o (iter, type)**: Remove the iterative training strategy and delete the entity type in the template of RE. In, this method, the template is "[CLS] *head entity* [SEP] *tail entity* [SEP] *sentence*".

The results of the ablation analysis are shown in Table 5. Compared to the complete framework with w/o (cumulative), it can be seen that the using of accumulated data through iterations is beneficial for improving the generalization ability of NER and RE models. The held-out performances of the model without iteration indicates that the iterative training strategy can not only discover the specific knowledge in the fine domain, but also maintain the ability to discover overlapping knowledge between coarse and fine domain. The RE performance of w/o (iter) is slightly better than that of w/o (iter, type), indicating that specifying the entity type of the entities is helpful for improving the performance of the RE task.

Table 4: Results of manual evaluations.

| models | $\mathbb{E}_{conf}$ | $\mathbb{T}_R$ | $\mathbb{T}_E$ |
|---|---|---|---|
| BERT | 0.78 | 0.50 | 0.66 |
| Bio_ClinicalBERT | **0.92** | 0.60 | 0.56 |
| biomed_RoBERTa | **0.92** | **0.70** | **0.72** |

Table 5: Results of ablation study.

| models | NER precision | RE precision |
|---|---|---|
| BERT | **0.878** | **0.977** |
| w/o (cumulative) | 0.845 | 0.970 |
| w/o (iter) | 0.857 | 0.967 |
| w/o (iter, type) | 0.857 | 0.967 |
| Bio_ClinicalBERT | **0.877** | **0.976** |
| w/o (cumulative) | 0.847 | 0.971 |
| w/o (iter) | 0.860 | 0.972 |
| w/o (iter, type) | 0.860 | 0.970 |
| biomed_RoBERTa | **0.871** | **0.973** |
| w/o (cumulative) | 0.822 | 0.967 |
| w/o (iter) | 0.858 | 0.971 |
| w/o (iter, type) | 0.857 | 0.968 |

## 5 CONCLUSION AND DISCUSSION

In this paper, we propose an integrated, end-to-end framework for knowledge graph domain adaptation using distant supervision, which can be used to construct KG from fully unlabeled raw text data with the guidance of an existing KG. To deal with the potential challenges in distant supervision, which might limit the knowledge discovered from the new domain, we propose an iterative training strategy, which divides an unlabeled corpus into multiple corpuses. For each new corpus to the model, we then combine the knowledge in the coarse domain with the knowledge identified from the previous corpuses for distantly-supervised training. By adopting the iterative training strategy, our proposed KGDA framework can discover not only knowledge that overlaps with the coarse domain, but also knowledge specific to the fine domain and unknown to the coarse domain, thus enabling coarse-to-fine domain adaptation. We implemented the adaptation from biomedical KG to the oncology domain in our experiments and verified the effectiveness of the KGDA framework through held-out and manual evaluation.

Several limitations and challenges remain beyond the current work for more effective and accurate KG construction: Firstly, more thorough evaluation with human reader study is needed to validate that new knowledge relevant (not only correct) to the target domain can be discovered by KGDA. Secondly, it has been recognized by the field that distant supervision will inevitably introduce noisy labels Liang et al. (2020); Zhang et al. (2021b), thus the denoising step is usually needed but not implemented in the current version of KGDA. Thirdly, there has been existing KG constructed in the related domains of oncology and cancer research. We will investigate the scheme to allow adaption from multiple sources (not only the coarse domain) to leverage this existing knowledge better. Another type of crucial prior information for this work is clinical ontology, where we will integrate the relationships defined in ontology and entity description to enhance the model. Fourthly, an essential premise of the KGDA is that we assume the source and target domains share the same set of entity types and relation types, which can limit the knowledge discovered from the fine domain. We will investigate data mining techniques to adaptively add/remove entity and relation types in the fine domain. Finally, there have been many new large-scale pre-trained language models developed such as GPT-3 in recent years. While our model uses variations of BERT (BiomedRoBERTa and BioClinicalBERT) as backbone networks, we can easily adapt KGDA to other language models.

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
