# OpenReview forum: "Coarse-to-fine Knowledge Graph Domain Adaptation based on Distantly-supervised Iterative Training"
_ICLR.cc/2023/Conference — Submitted to ICLR 2023_

### Official Review · Reviewer_8yCf · 2022-10-23

**Confidence:** 3
**Correctness:** 2
**Technical Novelty And Significance:** 1
**Empirical Novelty And Significance:** 1
**Recommendation:** 1

**Clarity, Quality, Novelty And Reproducibility:**

- Table 4: the table or caption should include the reported metric;
- Table 5: why precision instead of F1 is reported?


**Strength And Weaknesses:**

Strength:
- The task of learning finer-grained KG from an existing coarse KG without additional training data might be important.

Weaknesses:
1. The definition of domains is not clear. Why does biomedical KG not contain oncological information? Is the oncological domain defined by the 12 selected journal? How are these journals selected?
2. The related work section lacks descriptions of methods that tackle the same task (learning a finer-grained KG from an existing coarse KG);
3. The method section is lengthy (pages 3-7) and a little bit hard to follow. The authors should consider simplifying the notations or descriptions;
4. Most importantly, the evaluation is not valid: (1) the main comparison should be made with other work that tackles similar tasks, not just variants of different biomedical PLMs; (2) the manual evaluation consists of only 50 samples and 1 physician. This is far from enough; (3) the authors should prove and show that the proposed model can discover new knowledge in oncology. Specific case studies must be provided.

**Summary Of The Paper:**

This paper tackles the task of building a finer-grained KG from a coarse KG (biomedicine -> oncology). For this, the authors propose a pipeline that iteratively applies distant supervision to subsections of the fine domain corpus, which requires no annotated data for training. The authors compare the performance of 3 biomedical PLMs used in their pipeline and conduct a small-scale human evaluation.

**Summary Of The Review:**

This paper proposed a pipeline based on iterative distant supervision to learn a finer-grained KG from an existing coarse KG. The methods are not clearly presented, and the evaluation is invalid. I suggest rejecting this paper.

---

### Official Review · Reviewer_KkJe · 2022-10-24

**Confidence:** 5
**Clarity, Quality, Novelty And Reproducibility:** see above
**Correctness:** 3
**Technical Novelty And Significance:** 1
**Empirical Novelty And Significance:** 1
**Recommendation:** 3

**Strength And Weaknesses:**

Strengths:
1. The paper is well motivated and presented.
2. The proposed method is simple yet effective.

Weaknesses:
1. The novelty of the paper is limited, leaving the key challenges unsolved. For example, 1) how to ensure the newly discovered entities and relations are fine-grained, out of the given corse domain does not mean fine-grained. 2) How to avoid the error propagation during iterative training? 3) How to label the fine-grained types for newly discovered entities and relations?

2. The evaluation is not convincing. First, it lacks strong baseline. What if using distant supervision to annotate all corpus and applying the trained model on test set directly? Second, the improvements are marginal. Especially, in ablation study, the improvements from each component are very limited, what is the significance test results?

**Summary Of The Paper:**

The paper focuses on the poor annotation issue and proposes a distantly supervised iterative training method for knowledge graph construction. The authors leverage distant supervision to automatically label entities and triples from a find-grained domain based on a coarse domain. In specific, they first use distant supervision to annotate part of training corpus and train the NER and RE models. Then, the well-trained model will be applied on the left corpus to discover new entities and relations, where the high-confident ones will be used for further automate annotation. The experiments show improvements on the constructed test set. The presentation is mostly clear and easy to follow. However, there are several major issues that weaken the novelty.

**Summary Of The Review:**

see above

---

### Official Review · Reviewer_hrcf · 2022-10-24

**Confidence:** 3
**Correctness:** 3
**Technical Novelty And Significance:** 1
**Empirical Novelty And Significance:** 1
**Recommendation:** 3

**Clarity, Quality, Novelty And Reproducibility:**

The introduction should be carefully revised to highlight the motivation. Besides, the provided code is not easy to follow, which makes it hard to reproduce.

**Strength And Weaknesses:**

Strength:

I do not find any strength in this paper.

Weaknesses:

1.The proposed approach is not technically new and only utilizes existing methods for knowledge graph domain adaptation.

2.The experiments are also not conceivable without detailed analysis and case studies.

**Summary Of The Paper:**

This paper proposes an integrated framework for adapting and re-learning knowledge graphs from one coarse domain (biomedical) to a
finer-define domain (oncology). The proposed approach applies distant supervision on cross-domain knowledge graph adaptation. Consequently, no manual data annotation is required to train the model. This paper introduces a novel iterative training strategy to facilitate the discovery of domain-specific named entities and triples. Experimental results indicate that the proposed framework can perform domain adaptation and construction of knowledge graphs efficiently. Overall, this paper is not well-written and not novel at all.

**Summary Of The Review:**

I think this paper needs a major revision, and it benefits little for the ICLR community.

---

### Official Review · Reviewer_cH5Y · 2022-10-27

**Confidence:** 4
**Correctness:** 2
**Technical Novelty And Significance:** 1
**Empirical Novelty And Significance:** 1
**Recommendation:** 3

**Clarity, Quality, Novelty And Reproducibility:**

Please use the correct cite format. Use \citep instead of \cite.

The model is complicated and may be hard to reproduce.

**Strength And Weaknesses:**

The proposed system, if works, will serve as a powerful tool to construct domain-specific KBs.

The proposed system is fairly complicated. It's not clear whether the proposed system will also work for non-medical domain.

Another question comes from their evaluations. The performance of the proposed method is not compared to external baselines. It is not clear how challenging is the experiment task.

I am also concerned on the main focus of this paper. There's not much content on representation learning. It is also not clear how the proposed system will help other research in related fields.

**Summary Of The Paper:**

The paper proposed an accurate pipeline for constructing a domain specific database (fine). The constructed database is adapted from a coarse domain. The construction process is distantly (self) supervised without any manually annotated data.

**Summary Of The Review:**

Overall, this paper may be a useful for people who would like to build a domain-specific KB in low resource domains. However, it's not clear what are the learning components in this work. The results are not well evaluated (no external baselines). It is hard to evaluate the impact of this research.

---

### Decision · Program_Chairs · 2023-01-20

**Decision:**

Reject

**Justification For Why Not Higher Score:**

No technical novelty, problems with evaluation

**Justification For Why Not Lower Score:**

no lower score possible

**Metareview: Summary, Strengths And Weaknesses:**

SUMMARY

This paper adresses the task of building a finer-grained
knowledge graph (e.g., biomedicine) from a coarse knowledge
graph (e.g., oncology). The authors propose a pipeline that
iteratively applies distant supervision to subsections of
the special domain corpus. This requires no annotated data
for training. The authors compare the performance of three
biomedical PLMs on the task.  They also conduct a small
human evaluation.

STRENGTHS

Important problem addressed

WEAKNESSES

No technical novelty

No comparison to baselines

Manual analysis dataset too small

Discussion of related work not sufficient

Lack of analysis of experimental results

Key issues are not addressed, e.g., how to deal with
entities and relations that are newly discovered in the
target domain